# Early Intervention Including an Active Motor Component in Preterms with Varying Risks for Neuromotor Delay: A Systematic Review and Narrative Synthesis

**DOI:** 10.3390/jcm14041364

**Published:** 2025-02-18

**Authors:** Nele De Bruyn, Britta Hanssen, Lisa Mailleux, Christine Van den Broeck, Bieke Samijn

**Affiliations:** Department of Rehabilitation Sciences and Physiotherapy, Ghent University, 9000 Gent, Belgium; britta.hanssen@ugent.be (B.H.); lisa.mailleux@ugent.be (L.M.); christine.vandenbroeck@ugent.be (C.V.d.B.); bieke.samijn@ugent.be (B.S.)

**Keywords:** premature infant, early intervention, motor outcome, cerebral palsy

## Abstract

**Background/Objectives:** Previous reviews demonstrated stronger benefits of early interventions on cognition compared to motor outcome in preterm-born infants. Potentially, motor development needs more targeted interventions, including at least an active motor component. However, there is no overview focusing on such interventions in preterm-born infants, despite the increased risk for neuromotor delays. **Methods:** PubMed, Embase and Web of Science were systematically searched for (quasi-)randomized controlled trials regarding early interventions in preterm-born infants, with varying risks for neuromotor delay, and trials comprising an active motor component started within the first year were included. Study data and participant characteristics were extracted. The risk of bias was assessed with the Risk of Bias 2 tool. **Results:** Twenty-five reports, including twenty-one unique (quasi-)RCTs, were included and categorized as either pure motor-based interventions (*n* = 6) or family-centered interventions (*n* = 19). Of the motor-based interventions, four improved motor outcomes immediately after the intervention, and one of these also did so at follow-up, compared to five and one for family-centered approaches, respectively. Only five family-centered studies assessed long-term effects beyond age five, finding no greater efficacy than standard care. Overall, large variations were present for intervention intensity, type and outcomes between the included studies. **Conclusions:** Although methodological heterogeneity compromised conclusions, limited effects on motor outcome, in particular long-term outcome, were identified. Including a stronger motor-focused component embedded within a family-centered approach could potentially increase the impact on motor outcome, which would be of particular interest for infants showing early signs of neuromotor delay.

## 1. Introduction

Every year, an estimated 15 million babies are born worldwide before 37 weeks of gestation [1], 4.7% of them are born in Europe [2]. Although technical advances in neonatal care have increased the survival rate of extremely preterm infants [3], their vulnerability to neurodevelopmental disabilities remains a point of concern. An increasing number of studies examine the efficacy of early interventions within the first year of life with the aim of enhancing development and minimizing adverse outcomes. The importance of starting interventions early stems from the hypothesis of a benefit from the increased neural plasticity present during the first year of life [4]. The increasing number of studies on this topic resulted in Cochrane reviews with the most recent update published in January 2024 [5]. Here, the authors concluded that early interventions probably (low-certainty evidence) have a positive effect on cognitive and motor development during infancy, while high-certainty evidence was found for a more sustained effect up until preschool age, but only on the cognitive domain. Most early interventions encompass general developmental stimulation through parent education, enhancing parent–infant relationships and environmental enrichment. Potentially, motor development needs more targeted interventions, including at least an active motor component. Moreover, since thirty-six percent of very preterm-born infants present with motor difficulties at preschool age [6], a more delineated study of the efficacy of early interventions, including such an active motor component, on motor outcome is warranted. Such types of interventions could be of particular interest for infants showing early signs of neuromotor delay such as cerebral palsy (CP). CP is defined as a predominant sensorimotor disorder due to non-progressive lesions in the developing brain [7]. It is one of the most impairing outcomes after preterm birth, with a risk that increases exponentially with decreasing gestational age [6]. Another recent systematic review summarized interventions in infants and toddlers at high risk for or with a diagnosis of CP [8]. The authors reported very low-quality evidence for the effectiveness of task-specific motor training and constraint-induced movement therapy to improve motor function. However, this review included studies that provided therapy to participants up to the age of 32 months, which prevented the authors from drawing conclusions on the effect of early interventions within the first year of life. Moreover, although studied samples of infants at high risk for CP often encompass preterm-born infants, current reviews did not include both preterm and high-risk infants within their search strategy. Hence, the aim of this literature review is to provide an overview of studies in preterm-born infants with varying risks for neuromotor delay, reporting on the efficacy of early interventions, including an active motor component and starting within the first year of life.

## 2. Materials and Methods

This paper is written following the PRISMA statements for writing a systematic review [9] (see checklist in Appendix A). No pre-registration was performed for this study.

### 2.1. Research Question

What is the recent evidence for early intervention within the first year of life including an active motor component (I) for preterm-born infants with or without a high risk for CP (P) on motor outcome (O)?

### 2.2. Eligibility Criteria, Information Sources, Search Strategy

PubMed, Embase and Web of Science were searched for papers published between August 2015 and February 2024. Backward citation tracking was performed for all included papers after inclusion based on full text. The search strategy was built upon (P) preterm infants with or without a high risk for CP, (I) early intervention and (O) motor or developmental outcome. A full search strategy can be found in Appendix A. The inclusion and exclusion criteria are displayed in Table 1.

### 2.3. Selection Process

After the search was conducted, results were imported into Endnote to remove duplicates. Study selection was conducted independently by three reviewers (NDB, BS and BH) using Rayyan for screening of the title and abstract (round 1) and full text (round 2). Any disagreements were resolved through consensus via discussion. For interventions with unclear descriptions, clinical trial registrations or protocol papers were consulted whenever available.

### 2.4. Data Collection Process and Data Items

After full-text screening, tables of evidence were created by one reviewer (NDB) and checked by a second reviewer (LM). Thereby, studies were categorized into two groups based on their therapy content: 1) pure motor-based interventions and 2) general family-centered interventions. Studies were classified as pure motor-based in case their intervention focused predominantly on the improvement of motor development without incorporating components of a parent–infant relationship or parental education concerning other developmental domains. In contrast, in studies categorized as general family-centered interventions, parental education relating to the general development, behavior and parent–infant relationships played a key role. Any uncertainties were discussed with a second reviewer (CVDB or BH). See Appendix A for an overview of theoretical frameworks and content of intervention programs.

### 2.5. Risk of Bias Assessment

The Cochrane risk of bias tool 2 was used to assess the risk of bias [10], conducted by one reviewer (NDB), with consultations from a second reviewer in ambiguous or unclear cases. The randomization process, deviations from intended interventions, missing outcome data, measurement of the outcome and selection of reported results were assessed for potential bias. For the question “Were participants aware of their assigned intervention during the trial?”, it was assumed that infants were unaware of their assigned intervention due to their age. Regarding the question “Were carers and people delivering the interventions aware of participants’ assigned intervention during the trial?”, if parents/caregivers were involved in the intervention, it was interpreted as “yes” since they were aware of the intervention. Similarly, if healthcare professionals delivered the intervention directly to the child without parental involvement, it was interpreted from the perspective of the healthcare professional. Additionally, the question ‘allocation based on time of admission’ was rated as ‘random’ if allocation blocks were sufficiently long (several months) and age of inclusion was limited by gestational age, thereby preventing manipulation of allocation through delayed inclusion.

## 3. Results

### 3.1. Study Selection

The search strategy resulted in 154 PubMed, 373 Web of Science and 231 EMBASE articles. After duplicates were removed, 395 articles were screened on title and abstract, resulting in 82 articles for full-text screening and 21 for final inclusion. After manual screening of reference tables, we added another 4 articles. In total, we included 25 articles in this review (see Figure 1 for flow chart) [11,12,13,14,15,16,17,18,19,20,21,22,23,24,25,26,27,28,29,30,31,32,33,34,35]. These 25 articles covered 21 unique studies.

### 3.2. Study and Participant Characteristics

All motor-based studies (*n* = 6) were unique and included a randomized, evaluator-blinded design [11,12,13,14,15,16]. The final sample sizes for which results were reported ranged from 16 to 44 infants. The family-centered interventions (*n* = 19, [17,18,19,20,21,22,23,24,25,26,27,28,29,30,31,32,33,34,35]) also mainly involved studies with a randomized, evaluator-blinded design. Only one study used a non-blinded quasi-RCT [23], and one study did not report on blinding of the assessors [20]. The final sample sizes for which results were reported ranged from 11 to 242. Two RCTs, one conducted in the Netherlands [21,25,32] and one in Australia [29,30,31], each covered three publications included in this systematic review.

Overall, most studies were conducted in Western countries like the USA [11,13,20,22], Australia [24,29,30,31] and European nations [12,17,21,25,26,28,32,33], and one study included Italian and Danish participants [16]. Studies were further conducted in Brazil [14,15,23], South Korea [34], Switzerland [35] and Turkey [18,19,27]. A more detailed overview of the study and participant characteristics is provided in Table 2 and Table 3, respectively.

Risk of bias analysis using the Cochrane risk of bias tool 2 revealed only one study with low overall bias which was classified as a general family-centered intervention [35]. Except for two studies with high concerns [20,23], most other family-centered interventions were identified with some concerns. Of the motor-based interventions, two studies had some concerns [12,15], while four had high concerns [11,13,14,16]. Bias risks were primarily due to a lack of blinding during outcome measurement and the absence of predefined statistical plans for comparing final results.

### 3.3. Methodological Characteristics of the Interventions

#### 3.3.1. Motor-Based Interventions

A detailed overview of the methodological characteristics of the intervention can be found in Table 2 and Figure 2. Each study within the motor-based interventions consisted of a different therapy content including tethered kicking [11], crawling training with a mini-skateboard [12] or robotic system [13], reaching with [15] or without [14] a visual component or general motor stimulation [16]. As a control intervention, two studies used standard of care [15,16], two studies did not intervene in their control group [11,14] and two other studies used a control intervention with the same intensity [12,13]. Therapy intensity in the experimental group varied from one 4 min training [14] to 30–45 min practice daily for four weeks [16]. All interventions were delivered at home either by a therapist [12,14] or by a caregiver with support from a therapist at home [11,15] or remotely [16]. In one study, the therapist provider was not described [13]. The intervention started in all studies within the first 6.5 months corrected age. To assess intervention efficacy on motor outcome, three motor-based interventions included at least a standardized assessment for motor development such as the Infant Motor Profile (IMP), Test of Infant Motor Performance (TIMP), Alberta Infant Motor Scale (AIMS) or the Bayley Scales of Infant and Toddler Development (Bayley-III) [12,15,16]. The three other studies only reported intervention-specific assessments such as quantitative outcomes from the robotic crawling device and suit [13] or video recordings of tethered kicking [11] or reaching behavior [14]. In two studies, follow-up lasted until 12 months corrected age [11,12], while the other four studies only reported immediate post-intervention effects.

#### 3.3.2. Family-Centered Interventions

Within the general family-centered interventions, similar therapy concepts were often used across different studies, with COPCA [21,25,26,27,32,35], VIBeS Plus program (i.e., Victorian Infant Brain Studies) [29,30,31] and SPEEDI [22,24] being the most frequently used. As a control intervention, eleven studies used standard of care [17,22,23,24,28,29,30,31,33,34,35], and other studies used a control intervention with the same intensity [18,19,20,26,27] or traditional physiotherapy, the intensity of which varied depending on the pediatricians’ advice [21,25,32]. Regarding the experimental group, a large variation in therapy frequency and duration was found, with therapy intensity varying from 45–50 min per month over a six-month period [18] to 15–20 min per day during one year [20]. Most studies were delivered at home by the caregivers with support from a therapist. Three studies did not include home visits. In the study of Alberge et al., the therapy was delivered in a private practice by a psychomotor therapist [17], Ferreira et al. conducted their intervention program during standard follow-up assessments at the hospital [23], and Youn et al. provided group sessions by a pediatric physiotherapist in an outpatient center [34]. Four studies started the intervention already in the NICU before discharge [22,24,28,33]. Most other studies started the intervention within 3 to 4 months corrected age, except for three studies [17,19,26] that started at 9 to 10 months corrected age. To assess intervention efficacy on motor outcome, all studies that reported outcomes within the first two years of age included a motor developmental assessment using the IMP, TIMP, AIMS and/or Bayley-II or III. Studies beyond two years of age used the Developmental Coordination Disorder Questionnaire and the Vineland Adaptive Behavior Scales [25] or the Movement Assessment Battery for Children, second version (M-ABC-2) [29,30,31,33]. Follow-up lasted most frequently until 18 to 24 months corrected age. Only three studies focused solely on the immediate post-intervention effects [18,19,20], while five studies were focused on or included long-term outcomes beyond 2 years of age [25,29,30,31,33].

### 3.4. Intervention Effects

#### 3.4.1. Motor-Based Interventions

All motor-based interventions reported on the immediate post-intervention effects. Only the studies of Dumuids-Vernet et al. and Campbell et al. comprised a longitudinal follow-up analysis including follow-up assessments until 12 months corrected age. Dumuids-Vernet et al. found significantly better outcomes on the fine and gross motor scales of the Bayley-III after Crawli training compared to regular training in prone or no training immediately post-intervention and in their longitudinal follow-up analysis [12]. Positive effects were also reported by Sgandurra et al. [16], showing that the CareToy system was significantly better than standard care for outcomes on IMP and AIMS immediately after intervention. Kolobe et al. further reported that both reinforcement learning and error-based learning during crawling training on the robotic device resulted in a greater increase in arm and trial-and-error efforts compared to reinforcement learning alone [13]. Nascimento et al. found that reaching training with sticky mittens resulted in improved reaching compared to no training [14]. In contrast, Campbell et al. did not find an effect of tethered kicking training immediately after the intervention, nor in their longitudinal follow-up analysis, compared to no intervention [11]. Also, Rodovanski et al. did not find an effect of adding parental education on early stimulation targeting visual and motor functions to standard care on TIMP outcomes compared to standard care alone [15].

#### 3.4.2. Family-Centered Interventions

In the general family-centered interventions, mixed findings were found when compared to standard care. Van Hus et al. demonstrated longitudinal follow-up effects of the Infant Behavioral Assessment and Intervention Program on motor development (i.e., Bayley-II, M-ABC-2) across all time points [33]. Alberge et al. only found between-group differences in motor development (Bayley-III) at 9 m corrected age, but not at 24 m corrected age [17]. Two other studies were not able to demonstrate between-group differences immediately post-intervention or at follow-up [28,34], while Ferreira et al. found higher scores for gross motor development on Bayley-III immediately post-intervention in favor of the control group [23].

Three studies compared their early intervention program with a control group of equal intensity and did not reveal significant between-group differences for early motor development immediately post-intervention [18,19,20] or at follow-up [18]. Yet, Apaydin et al. additionally reported within-group improvements for motor development on Bayley-III only in the experimental group [19].

Significant differences in favor of COPCA compared to traditional infant physiotherapy were demonstrated by Dirks et al., with infants receiving COPCA being more often bathed in a sitting position at 6 m corrected age [21], and by Ziegler et al. showing better scores on the variation and performance domain of the IMP at 18 m corrected age [35]. Although no further between-group differences were found for postural control before 18 m corrected age [32], early motor development assessed with the IMP and Bayley II [26] or Bayley III [27,35] or motor outcome at primary school age (7.5 to 10 years, M-ABC-2) [25], within-group differences for both groups were present, suggesting improved postural control [32] and fine and gross motor development on Bayley-III [27] independent of group allocation.

Varying results were found for SPEEDI compared to standard care. Dusing et al. demonstrated within-group improvements, but no between-group differences in reaching skill or early motor development (TIMP, Bayley-III) [22]. In contrast, Finlayson et al. found significantly fewer infants with absent fidgety movements at 3 m corrected age and better gross motor scores on Bayley-III at 4 m corrected age in favor of SPEEDI compared to standard care [24].

From the VIBeS Plus program, only publications with long-term follow-up were included [29,30,31]. In this RCT, no evidence was found for an effect on motor outcome, as motor scores on the M-ABC-2 were similar between the intervention group and standard care group at 8-year [29] and 13-year [31] follow-up.

#### 3.4.3. Covariates

Interestingly, early interventions seem to be more beneficial for infants born from mothers with a lower educational level or in families with higher social risks. The VIBeS plus program reported that 8- and 13-year-old children with higher social risk have better motor scores (M-ABC-2) after the intervention program compared to the standard care group, while this was not the case for children with lower social risk [30,31]. Also, Alberge et al. found a stronger effect of their psychomotor therapy on fine motor scores (Bayley-III) in infants from mothers with a lower educational level [17]. However, Van Hus et al. did not find a clear intervention effect of maternal education [33]. In contrast, these authors reported that infants with bronchopulmonary dysplasia benefited more from the early intervention, compared to infants without bronchopulmonary dysplasia. Finally, Cooper et al. showed that gestational age and small for gestational age were correlated with the TIMP change scores (from 0 to 3 months corrected age) during the first 3 months of their one-year intervention, indicating that a higher gestational age or birth weight was associated with a higher increase in TIMP scores [20]. (A detailed overview of the results for each study can be found in Appendix A.)

## 4. Discussion

This systematic review provided an overview of the evidence on the effectiveness of early interventions with an active motor component started in the first year of life in preterm-born infants with varying risks for neuromotor delays. We included 25 studies, including 21 unique (quasi-)RCTs, which were categorized as either pure motor-based interventions (*n* = 6) or family-centered interventions (*n* = 19). Four pure motor-based interventions revealed improved motor outcomes immediately post-intervention, one of which also did so at follow-up, while for the family-centered interventions, this was the case for only five studies and one study, respectively. Subsequently, 2 motor-based interventions and 11 family-centered interventions did not identify motor improvements immediately after the intervention, compared to a control group of equal intensity of traditional physiotherapy or standard care. Moreover, long-term effects beyond the age of five years of such early intervention programs were studied only by five family-centered interventions and were not deemed more efficacious compared to standard care [25,29,30,31,33]. The lack of significant findings could potentially be explained by the fact that standard care and traditional physiotherapy are often already at a high level. Moreover, while the transfer to daily life activities may be more limited in the motor-based interventions [36] due to the lack of providing challenging environments for the infant in daily life or focusing on one specific motor skill (i.e., crawling, reaching or kicking) [11,12,13,14,15], motor development might still not be targeted specifically enough within the family-centered interventions. We could hypothesize that for improving motor development, a more direct approach is required, like in the motor-based interventions, rather than general motor stimulation embedded in a broader developmental stimulation program in the family-centered interventions. If so, this could be of particular interest for infants who are at high risk for CP or who show early signs of CP. Indeed, in their meta-analysis, Baker et al. concluded that task-specific motor training improves motor function in infants and toddlers with CP [8]. However, the conclusions of Baker and colleagues were based on low-quality evidence, urging the need for high-level RCTs.

Moreover, CP is a condition that does not solely affect motor outcome [7]. Also, preterm-born infants in general have an increased risk for developmental problems beyond the motor domain [6]. Hence, providing purely motor-based interventions would not serve these other developmental concerns, and we know that high-certainty evidence is already present for the positive impact of early interventions on cognitive outcome in infants born preterm. Consequently, it would be of interest to study CP-specific outcomes on the efficacy of targeted motor interventions embedded within a family-centered intervention and find a new balance between both. On the one hand, we know that based on the principle of motor learning and experience-dependent neuroplasticity [37], training should be specific. In this way, the motor component will boost the motor skills in a short time, when age matters and brain plasticity is still high, underlining the importance of a targeted motor component for improving motor outcomes. On the other hand, it is already well known that parents play an indisputable role within early interventions [5]. Parents will learn how to create stimulating trial-and-error experiences which will help the transfer of learned skills to activities of daily living. Parents will acquire the knowledge of implementing these principles, facilitating their sustained implementation well beyond the intervention time frame, thereby extending the impact.

Furthermore, new interventions should study if long-term effects would be enhanced by implementing boost programs. So far, long-term motor outcomes beyond the age of five years have not shown an effect [25,29,30,31,33]. However, these studies evaluated the effect of one intervention period. We hypothesize that the mechanisms of experienced-dependent neuroplasticity, which would assume restructuring of the brain in the first year of life, are also the basis of a long-term effect, indicating the need to maintain stimulation of the acquired neural pathways to keep them active and effective [37]. Another hypothesis could be that the neural groups responsible for early motor milestones and trained with the early interventions differentiate from the neural groups needed for more high-level motor skills which are often shown delayed at older ages [38].

Future research should also take into account that personal and environmental aspects could be determining factors influencing the (long-term) effects of early intervention, such as the social risk profile [30,31]. The inclusion of CP-related outcomes to evaluate therapies specifically for CP and better identification of infant and family characteristics that influence treatment outcomes would aid in delineating which kind of early intervention the infant will benefit from the most, and result in improving patient selection and cost-effectiveness and decreasing waiting times for starting specific early interventions.

However, this systematic review also has some limitations. At the study level, only a few studies included the interaction between group and time in their statistical analyses, which is needed to examine if there are differences between groups over time. Additionally, preterm birth contains still a rather large range of gestational ages with differences in expected outcomes, and more specific CP-related outcomes were often lacking even in the studies focusing on infants with a high risk for CP. Furthermore, specifically for the family-centered interventions, we were not able to extract the amount of stimulation parents had attributed to the motor development compared to other developmental domains from the articles. Hence, this needs to be taken into account when interpreting the results. At the review level, we need to acknowledge the date restriction, since we have searched for eligible articles from 2015 onwards. We aimed to focus on studies in the past decade due to the improved neonatal care, in particular for non-invasive respiratory support, coinciding with an increase in survival to discharge and decrease in morbidity [39]. Also, articles published after February 2024 were not included. A quick (not systematic) search already showed three new interesting papers within the scope of this review [40,41,42]. Nevertheless, this date restriction needs to be taken into account when interpreting the results. Moreover, we need to acknowledge that the inclusion criteria (at least 50% of the infants being preterm) led to the exclusion of some relevant studies such as BabyCIMT [43,44], and that the content of the intervention can only be interpreted based upon published information from the authors. It is possible that our interpretations of interventions may not reflect what has occurred due to a lack of detail in the papers. Also, the pre-registration of the protocol and performance of a meta-analysis and sensitivity analysis would have strengthened the conclusions of this review. However, methodological heterogeneity was very high, including a high amount of differences between intervention method, intensity and frequency, in outcome measures as well as the kind of scores (scaled score, composite score, raw scores) reported. Moreover, often it was not clear whether the improvements could be attributed to more than just spontaneous evolution or maturation of motor function. In particular, since all except one study [14] investigated the effect of an intervention over several weeks, spontaneous maturation of motor function resulting in better motor scores can be expected. Some studies have countered this by reporting the percentile scores calculated for specific age ranges.

Nevertheless, this review provided a detailed overview of the efficacy of early interventions including an active motor component in preterm-born infants with varying risks for neuromotor delays, further improving our knowledge of such early interventions on motor development and providing directions for future research.

## 5. Conclusions

This systematic review provided an overview of the evidence on the effectiveness of early interventions with an active motor component in preterm-born infants with varying risks for neuromotor delay, which were classified as either pure motor-based interventions or family-centered interventions. Although methodological heterogeneity between studies compromised study conclusions and clinical implications, we identified limited effects of these early interventions on motor outcome, in particular for long-term follow-up. Acknowledging the importance of family-centered interventions for other outcomes, we hypothesize that including a stronger motor-focused component within a family-centered approach would increase the impact on motor outcome, which would be of particular interest for infants at high risk for CP.

## Figures and Tables

**Figure 1 jcm-14-01364-f001:**
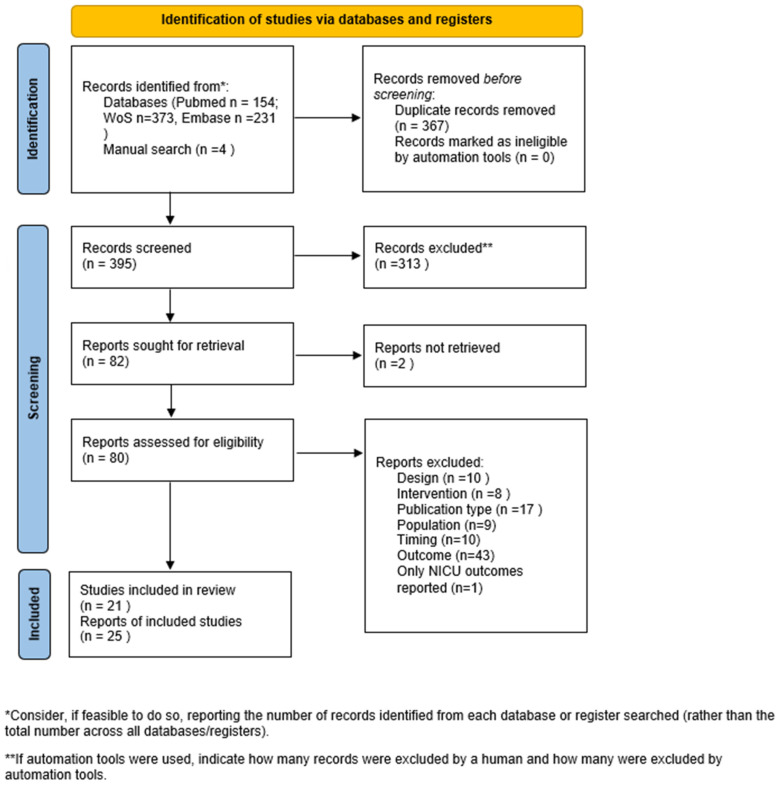
Flow chart of study inclusion. NICU: Neonatal Intensive Care Unit, *n*: number.

**Figure 2 jcm-14-01364-f002:**
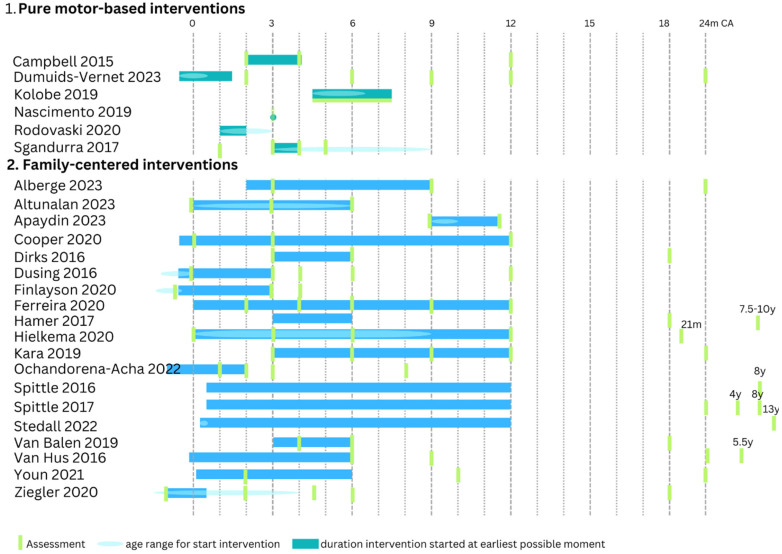
Overview of (1) pure motor-based interventions and (2) family-centered interventions. For the age to start the intervention, a range of ages was often used; this is indicated with a light blue ellipse. When a study started at a fixed age, no ellipse was drawn. The duration of an intervention is indicated with a bar assuming the earliest possible moment of starting. When the intervention starts at an older age (within the age range for the start of the intervention), the entire bar shifts towards the new starting age [11,12,13,14,15,16,17,18,19,20,21,22,23,24,25,26,27,28,29,30,31,32,33,34,35]. CA = corrected age, y = year, m = months.

**Table 1 jcm-14-01364-t001:** Eligibility criteria.

	Inclusion Criteria	Exclusion Criteria
Population	Preterm-born infants with or without a high risk for cerebral palsy of which at least 50% were born <37 weeks or <1500 g birthweight, or preterm infants should be analyzed separately	Infants with specific syndromes; More than 50% of participants were born >37 weeks and >1500 g BW
Intervention	What: Stimulation motor function, developmental stimulation; therapy needs to contain an active component Who: Given by professional or given by parents who are educated and coached by professional When: Started at hospital and continued at home or started after hospital discharge within 12 months CA	What: Medical; drugs; stress reduction; feeding/milk When: Only provided at NICU; started later than 12 months after birth
Comparison	Standard care, other intervention	
Outcome	Motor function CP; DCD; child development; psychomotor function	Birthweight; stress reduction; cognitive or social development
Study design	RCT, quasi-randomized allocation	Case–control, systematic review, meta-analysis; case study, cohort study
Language	English, Dutch, French	Other
Full text available	Available online or by contacting corresponding author	Not available
Publication date	August 2015–February 2024	Before August 2015

Abbreviations: g, grams; BW, birth weight; NICU, neonatal intensive care unit; CP, cerebral palsy; DCD, developmental coordination disorder; RCT, randomized controlled trial.

**Table 2 jcm-14-01364-t002:** Study characteristics.

(A) Study Characteristics of the Motor-Based Interventions								
Author	ROB	Study Design	Intervention Groups	Analyzed Sample Size	Treatment Delivery	Frequency/Intensity	Motor Outcome Measures	Timepoints
Campbell et al., 2015 [11]		Longitudinal pilot study with random allocation and blinded evaluators	Tethered kicking	*n* = 7	Delivered at home by the parents, supervised by a physical therapist	8 min/d; 5 d/w from 2 to 4 months CA	Video of tethered kicking * Head movements Hip flexion/extension	2, 4 and 12 months CA
			No intervention	*n* = 9	NA	NA		
Dumuids-Vernet et al., 2023 [12]		Evaluator-blinded RCT	Crawling training on the Crawliskate	*n* = 15	Delivered at home by an osteopath	5 min/d; 7 d/w; 8 weeks	**Bayley-III** Prechtl’s GMA ATNAT ASQ-3	2, 6, 9 and 12 months CA
			Prone positioning on mattress	*n* = 14	Delivered at home by an osteopath	5 min/d; 7 d/w; 8 weeks		
			Standard care	*n* = 15	NI	NI		
Kolobe 2019 [13]		Repeated measures experimental design with random allocation and blinded evaluators	Reinforcement learning and error-based learning	*n* = 14	Delivered at home	3 × 5 min; 2 d/w; 12 weeks	Activity recognition sensor suit pathlength transversed by several points of the body * Movement Observation Coding Scheme	Each session
			Reinforcement learning only	*n* = 9	Delivered at home	3 × 5 min; 2 d/w; 12 weeks		
Nascimento et al., 2019 [14]		Evaluator-blinded RCT	Sticky mittens with open fingers	*n* = 12	Delivered at home by a researcher	Once 4 min	Reaching protocol (**reaching enhancement**)	Immediately before, after and 4 min after training
			Spontaneous limb movements	*n* = 12	Delivered at home by a researcher	Once 4 min		
Rodovanski et al., 2021 [15]		Evaluator-blinded RCT	Standard care with additional information about early stimulation targeting visual and motor functions	*n* = 14	Delivered at home by a caregiver, supported by a researcher	10–15 min; 7 d/w; 28 days	TIMP *	One week before (T0) intervention and 1–4 days after (T1) intervention
			Standard care	*n* = 16	Parental education using illustrated handbook	One contact moment		
Sgandurra et al., 2017 [16]		Evaluator-blinded RCT	CareToy System	*n* = 19	Delivered at home by a caregiver, supported by rehabilitation staff	30–45 min; 7 d/w; 4 weeks	**IMP** AIMS	Before (T0) and after intervention (T1), T2 after changing treatment groups, T3 at 18 months CA.
			Standard care	*n* = 22	Parental education	Bimonthly follow-up checks; PT if necessary		
**(B) Study Characteristics of the Family-Centered Interventions**								
**Author**	**ROB**	**Study Design**	**Intervention Groups**	**Sample Size**	**Treatment Delivery**	**Frequency/Intensity**	**Motor Outcome Measures**	**Timepoints**
Alberge et al., 2023 [17]		Evaluator-blinded RCT	Psychomotor therapy	*n* = 57	Delivered at private practice by a psychomotor therapist	20 1 h sessions, 1×/week for 4 months and then every 15 days for the next 4 months discontinued at 9 months	**Bayley-III** Neurological examination	9, 24 months
			Standard care	*n* = 57	NI	NI		
Altunalan et al., 2023 [18]		Evaluator-blinded RCT	Explorer Baby	*n* = 28	Delivered by experienced therapist + home program	45–50 min/month during 6 months	**Bayley-III**	Before (T0), during (T1) and after therapy (T2)
			NDT	*n* = 29	Delivered by experienced therapist + home program	45–50 min/month during 6 months		
Apaydin et al., 2023 [19]		Evaluator-blinded RCT	SAFE early intervention	*n* = 12	Delivered at home by a parent supported by a therapist	Every day practice for 10 weeks (intensity documented via logbook)	Bayley-III COPM *	Baseline (T1) and 10 weeks later (T2)
			NDT	*n* = 12	Delivered at home by a parent supported by a therapist	Every day practice for 10 weeks (intensity documented via logbook)		
Cooper et al., 2020 [20]		RCT (no information about blinding evaluators)	Assisted exercise (based on NDT) and enhanced social interaction program	*n* = 48	Delivered at home by a parent supported by a therapist	At least 15–20 min/day during one year	TIMP: T0, T1 AIMS: T1, T2 #	At NICU discharge (T0), 3 months CA (T1) and 1 year CA (T2)
			Enhanced social interaction alone	*n* = 51	Delivered at home by a parent supported by a therapist	At least 15–20 min/day during one year		
Dirks et al., 2016 [21]		Evaluator-blinded RCT	COPCA	*n* = 18	Delivered at home by a COPCA therapist	1 h, 2×/w; 3 months	PEDI (functional mobility) (**bathing position)**	18 months
			Traditional infant physiotherapy (mainly based on NDT)	*n* = 21	Mostly delivered at home by a pediatric physiotherapist	The frequency varied from 2 to 28 times, and the duration from 12 to 50 min		
Dusing et al., 2018 [22]		Evaluator-blinded RCT	SPEEDI + standard care	*n* = 5	Phase 1 (parental education) delivered in NICU, phase 2 at home by a parent supported by a SPEEDI therapist	Phase 1: 21 days Phase 2: 20 min/d; 5 d/w; 12 weeks	**Reaching skill:** T2, T3, T4 TIMP: T0, T1, T2, T3 BAYLEY-III: T4	T0: baseline T1: end phase 1 (21 days after baseline), T2: end phase 2 (12 weeks after end Phase 1), T3: follow-up 1 (1 month after end phase 1) T4: follow-up 2 (2 months after follow-up 1 or 3 months after end phase 2)
			Standard care	*n* = 6	Referral to therapy services in the NICU and to their local Early Intervention program	NI		
Ferreira et al., 2020 [23]		A non-blinded quasi-experimental RCT	Standard care + early intervention	*n* = 72	Delivered at the hospital during follow-up	5 follow-up assessments + 1 h early intervention	**Bayley-III**	2, 4, 6, 9 and 12 months CA
			Standard care	*n* = 170	Parental education; professional referral if required	5 follow-up assessments		
Finlayson et al., 2020 [24]		Evaluator-blinded RCT	SPEEDI + standard care	*n* = 8	Phase 1 (only parental education) delivered in NICU, phase 2 at home by a parent supported by a SPEEDI therapist	Phase 1: 21 days Phase 2: 20 min/d; 5 d/w; 12 weeks	GMA: T0, T1 TIMP: T0, T1, T2 Bayley-III: T2 #	T0: baseline (between 34 and 38 + 6 GA) T1: 3 months CA T2: 4 months CA
			Standard care	*n* = 9	Parental education; developmental follow-up services after discharge	NI		
Hamer et al., 2017 [25]		Evaluator-blinded RCT	COPCA	*n* = 18	Delivered at home by a COPCA therapist	1 h, 2×/w; 3 months	**VABS** DCD-Q	Between 7.5 and 10 years
			Traditional infant physiotherapy (mainly based on NDT)	*n* = 22	Mostly delivered at home by a pediatric physiotherapist	NI		
Hielkema et al., 2020 [26]		Evaluator-blinded RCT	COPCA	*n* = 23	Delivered at home by a COPCA therapist	30–60 min; 1×/w; 1 year	**IMP**: T0, T1, T2, T3, T4 AIMS: T0, T1, T2, T3, T4 Bayley-III: T0, T1, T2, T3, T4 GMFM: T0, T1, T2, T3, T4	T0: baseline T1: after 3 months T2: after 6 months T3: after 12 months T4: 21 CA
			Traditional infant physiotherapy	*n* = 20	Mostly delivered at home by a pediatric physiotherapist	30–60 min; 1×/w; 1 year		
Kara et al., 2019 [27]		Evaluator-blinded RCT	Family-based intervention based on COPCA	*n* = 16	Delivered at home by a parent with coaching of a COPCA therapist	60 min; 2×/w; 9 months	**Bayley-III**	3, 6, 9, 12 and 24 months CA
			Traditional early intervention	*n* = 16	Delivered by an experienced pediatric physiotherapist	60 min; 2×/w; 9 months		
Ochandorena-Acha et al., 2022 [28]		Evaluator-blinded RCT	Early physiotherapy intervention program + standard care	*n* = 20	Delivered at NICU and continued at home by the parents supported by a physiotherapist	NICU: 15 min; 2×/d; 10 days Home: 15–20 min; 2×/d; 5 d/w until 2 months CA	**AIMS** ASQ-3	T1: 1 or 2 months CA T2: 8 months CA
			Standard care (NIDCAP)	*n* = 21	Sporadic physiotherapy sessions if required	NI		
Spittle et al., 2016 [29]		Evaluator-blinded RCT	Preventative care program (VIBeS program) + standard care	*n* = 53	9 home visits by a physiotherapist and psychologist	90–120 min at 2 w, 4 w, 3, 4, 6, 8, 9 and 11 months CA	M-ABC-2 *	8-year follow-up
			Standard care	*n* = 47	Medical and developmental follow-up	NI		
Spittle et al., 2018 [30]		Evaluator-blinded RCT	Preventative care program (VIBeS program) + standard care	High/low social risk: *n* = 19/39	9 home visits by a physiotherapist and psychologist	90–120 min at 2 w, 4 w, 3, 4, 6, 8, 9 and 11 months CA	**Bayley-III** and M-ABC-2 in relation to social risk of the family	2-year, 4-year and 8-year follow-up
			Standard care	High/low social risk: *n* = 25/32	Medical and developmental follow-up	NI		
Stedall et al., 2022 [31]		Evaluator-blinded RCT	Preventative care program (VIBeS program) + standard care	*n* = 43	9 home visits by a physiotherapist and psychologist	90–120 min at 2 w, 4 w, 3, 4, 6, 8, 9 and 11 months CA	M-ABC-2 *	13-year follow-up
			Standard care	*n* = 38	Medical and developmental follow-up	NI		
Van Balen et al., 2019 [32]		Evaluator-blinded RCT	COPCA	*n* = 21	Delivered at home by a COPCA therapist	1 h, 2×/w; 3 months	Postural control assessed with EMG *	4, 6 and 18 months CA
			Traditional infant physiotherapy (mainly based on NDT)	*n* = 25	Mostly delivered at home by a pediatric physiotherapist	Depended on pediatrician’s advice		
Van Hus et al., 2016 [33]		Evaluator-blinded RCT	Infant Behavioral Assessment and Intervention Program (IBAIP)	*n* = 86	Mostly delivered at home by an IBAIP therapist	One intervention before discharge; 6 to 8 1 h sessions at home	Bayley-II M-ABC-2 *	6, 12 and 24 months CA 5.5 years
			Standard care	*n* = 85	NI	NI		
Youn et al., 2021 [34]		Evaluator-blinded RCT	Preventive intervention program	*n* = 69	Group sessions delivered at an outpatient center by a pediatric physiotherapist	4 home visits by nurse for understanding behavioral cues until 2 months CA; between 3 to 6 months CA, 12 neurodevelopmental group sessions of 90 min	**Bayley-III** Korean Developmental Screening Test	10 and 24 months CA
			Standard care	*n* = 67	NI	NI		
Ziegler et al., 2021 [35]		Evaluator-blinded RCT	COPCA	*n* = 8	Delivered at home by a parent with coaching of a COPCA therapist	30–45 min; 1×/w; 6 months	**IMP** PEDI Bayley-III Neurological examination	Baseline, 3 and 6 months after baseline and 18 months CA 2 years CA
			Standard care	*n* = 8	Mostly delivered in an outpatient setting by a pediatric physiotherapist	The frequency of the sessions varied from 11 to 30 times, and their duration from 28 to 40 min		

Abbreviations: ROB, risk of bias analysis with green for low overall bias, orange for some concerns and red for high overall bias; NDT, neurodevelopmental treatment; SAFE, Sensory strategies, Activity-based motor training, Family collaboration, and Environmental Enrichment; COPCA, COPing and CAring for infants with special needs; SPEEDI, Supporting Play Exploration and Early Developmental Intervention; NIDCAP: the Newborn Individualized Developmental Care and Assessment Program; VIBeS, Victorian Infant Brain Studies; IBAIP, Infant Behavioral Assessment and Intervention Program; *n*, number; NA, not applicable; NI, no information; min, minutes; d, days; w, week;h, hour; CA, corrected age; Bayley-III: Bayley scale of infant development third version; TIMP, Test of Infant Motor Performance; AIMS, Alberta Infant Motor Scale; GMA, general movements assessment; ATNAT, Amiel-Tison Neurological Assessment; ASQ, Ages and Stages Questionnaire; IMP, Infant Motor Profile; COMP, Canadian Occupational Performance Measure; PEDI, Pediatric Evaluation of Disability Inventory; VABS, Vineland Adaptive Behavior Scales; DCD-Q, Developmental Coordination Disorder Questionnaire; GMFM, Gross Motor Function Measure; M-ABC-2, Movement Assessment Battery for Children, 2nd version; EMG, electromyography. **Primary outcome measure** is indicated in bold, **#**, primary outcome measure was not motor-related, * no primary outcome measure mentioned in the paper or available protocol; RCT: randomized controlled Trial; NICU, neonatal intensive care unit;

**Table 3 jcm-14-01364-t003:** Participant characteristics.

(A) Participant Characteristics of the Motor-Based Interventions							
Author	Mean or Median Gestational Age (Range)		Mean or Median Birth Weight (Range)		Age Start Intervention	Inclusion Criteria	Exclusion Criteria
	EG	CG	EG	CG			
Campbell et al., 2015 [11]	22.4 w (23–30)	28.1 w (24–32)	NI	NI	2 months CA	Grade III or IV IVH or PVL Healthy enough to start exercise program at discharge	/
Dumuids-vernet et al., 2023 [12]	29 w (NI)	29 w (NI)	1202 g (NI)	1294 g/ 1227 g (NI)	Between 37 and 42 weeks of GA	24–32 w GA; living within 10 km of the laboratory; able to leave NICU and begin training between 37 and 42 w GA	Major brain damage; hypoxic–ischemic encephalopathy; congenital anomalies; bronchopulmonary dysplasia with oxygen dependence after 36 w GA; digestive or other problems preventing prone positioning; limb deformities; presence of retinopathy or sensory pathology that may delay motor development
Kolobe et al., 2019 [13]	<32 w: *n* = 7 32–37 w: *n* = 1 >37 w: *n* = 6	<32 w: *n* = 3 32–37 w: *n* = 3 >37 w: *n* = 3	NI	NI	Between 4.5 to 6.5 months of age	TIMP z score < −1.0; confirmed diagnosis of CP or positive MRI result	Congenital deformities of bones or joints; uuncontrolled seizures
Nascimento et al., 2019 [14]	35.86 w (NI)	36.08 w (NI)	2530 g (NI)	2770 g (NI)	4 months (12 weeks CA)	Late preterm infants Pre-reaching infants: 3–5 goal-directed reaches performed within 1 min	Anoxia; signs of neurological complications; hyperbilirubinemia; congenital malformations; syndromes; sensory dysfunction; cardiopulmonary difficulties; growth restriction; adequate weight for GA; <3 goal-directed reaches performed within 1 min.
Rodovanski et al., 2020 [15]	36 w (34–36)	36 w (35–37)	2685 g (1900–3125)	2905 g (2395–3535)	Between 30 and 89 days postnatally	Low-risk preterm infants; GA 28–37 w; age at enrolment between one to two months CA; absence of visual impairments according to the Red Reflex Examination and complete ophthalmologic exam; delayed visual tracking	Infants diagnosed with neurological or respiratory diseases; signs of hypoxemia, hyperventilation or hypo-ventilation during assessments; presence of congenital diseases; visual impairments; extreme prematurity; birth weight < 1000 g; unstable physiological conditions; any kind of intervention such as physical therapy, occupational therapy, early intervention, aquatic stimulation, at the same time that our stimulation protocol is being applied; infants with medical fragility that prevented them from participating
Sgandurra et al., 2017 [16]	30.7 (NI)	30.82 (NI)	1368 g (NI)	1459 g (NI)	Between 3 and 5.9 months CA	Preterm infants with GA between 28–32 w + 6; aged 3–9 months of CA who had achieved a predefined cut-off score in gross motor ability derived from ASQ-3	Birth weight < pc10, brain damage; any form of seizures; severe sensory deficits; other severe non-neurological malformations; participation in other experimental rehabilitation studies
**(B) Participant Characteristics of the Family-Centered Early Intervention**							
**Author**	**Mean or Median Gestational Age (Range)**		**Mean or Median Birth Weight (Range)**		**Age Start Intervention**	**Inclusion Criteria**	**Exclusion Criteria**
	**EG**	**CG**	**EG**	**CG**			
Alberge 2023 [17]	27.6 w (NI)	27.4 w (NI)	1056 g (NI)	1069 g (NI)	Between 2 and 9 months following hospital discharge	GA 24–29 w Hospitalized at Toulouse University Children’s Hospital	Alive at 34 w GA; congenital malformations; genetic diseases; IVH grade III-IV; cystic PVL; infants whose mother has a documented psychotic illness; families not speaking French
Altunalan 2023 [18]	29 (NI)	29 (NI)	1204 g (NI)	1321 g (NI)	NI	<33 w GA <6 m CA at enrolment	High risk for CP (based on neurological examination grades II, III, IV cranial imaging, cramped synchronized or absent fidgety movements); metabolic or genetic diseases; parents not speaking Turkish; parents with psychiatric diagnoses
Apaydin 2023 [19]	29.2 w (26–34)	31.1 (26–34)	1331 g (NI)	1653 g (NI)	Between 9 and 10 months CA	<37 w GA; NICU stay > 15 days; CA 9 to 10 months; willingness to participate	Congenital anomaly or systemic diseases; medical conditions which prevent active participation in therapy (such as oxygen dependence); living out of reach of the research team for home visits
Cooper 2020 [20]	27 (NI)	27 (NI)	933 (NI)	930 g (NI)	Within 2 weeks post-discharge	Healthy preterm infants, unlikely to develop serious complications; caregiver > 18 y GA < 29 w; GA at recruitment: >34 w on full feeds and nearing discharge	Significant lung disease requiring oxygen or corticosteroids at discharge; IVH gr III-IV; necrotizing enterocolitis; tracheostomy; bone diseases; skin disorders; symptomatic congenital heart disease; other congenital anomalies likely to severely impact the ability of the premature baby to participate
Dirks et al., 2016 [21]	29 w (27–40)	30 w (25–39)	1195 g (585–4750)	1190 g (635–3460)	3 months CA	Infants admitted to NICU with abnormal GMA around 10 w CA	Severe congenital anomalies; parents with insufficient understanding of Dutch
Dusing et al., 2016 [22]	25 w (NI)	26 w (NI)	840 g (NI)	680 g (NI)	35 to 40 weeks of GA, when medically stable	GA < 29 w AND/OR neonatal diagnosis of brain injury (IVH gr III-IV; periventricular white matter injury, HIE, hydrocephalus requiring shunt; living within <30 min of hospital; English-speaking	Diagnosis of genetic syndrome; musculoskeletal deformity
Ferreira et al., 2020 [23]	NI	NI	NI	NI	2 months CA	All newborns who remained at least three days at a reference maternity hospital located in a poor neighborhood or born at a University Hospital with a first developmental follow-up appointment during the recruitment period	Evident signs of neurological or sensory impairments; infants whose mother died at birth or had mental disorders that comprised their understanding of the research procedures
Finlayson et al., 2020 [24]	26.05 w (NI)	27.19 w (NI)	795.25 g (NI)	842.33 g (NI)	35 to 40 weeks of GA	Born ≤30 w GA; one English-speaking parent; living within 30 km of the hospital; medically stable and of ventilator support at enrolment. Off ventilator support at enrolment	Genetic syndrome or musculoskeletal deformity that could affect development
Hamer et al., 2017 [25]	30 w (27–40)	30 w (26–39)	1415 g (670–4750)	1205 g (635–3460)	3 months CA	Infants admitted to NICU with abnormal GMA around 10 w CA	Congenital anomalies; caregivers with inappropriate understanding of the Dutch language
Hielkema et al., 2020 [26]	32 w (26–41)	29 w (26–41)	1915 g (770–4410)	1375 g (720–5400)	Between 0 and 9 months CA	0–9 m CA with a vvery high risk of CP through: cystic PVL; parenchymal lesions following infarction or haemorrhage; severe asphyxia with brain lesions on MRI; clinical dysfunction suspect for development of CP	Insufficient understanding of the Dutch language; severe congenital anomalies
Kara et al., 2019 [27]	28.85 w (26.43–32)	29 w (27–32)	1285 g (710–1500)	1360 g (920–1500)	3 months CA	BW ≤ 1500 g; 3 months CA; abnormal GM’s at fidgety age	Congenital malformations or epilepsy; undergone multiple surgeries; no consent
Ochandorena-acha et al., 2022 [28]	31.84 w (NI)	32.05 w (NI)	1462.46 g (NI)	1590.79 g (NI)	After 32 w GA and before term-equivalent age when medically stable	Born between 28 + 0 and 34 + 0 weeks GA; parents stayed at the hospital > 6 h/day and able to speak and understand Spanish	Preterm triplets; major central nervous system injury (gr III-IV IVH or PVL); severe musculoskeletal or congenital abnormalities; BPD; major surgery; severe sepsis; necrotizing enterocolitis during neonatal period; hearing impairment; retinopathy due to prematurity; infants of mothers with documented history of social problems or mental illness
Spittle et al., 2016 [29]	27.4 w (NI)	27.5 w (NI)	1062 g (NI)	1018 g (NI)	2 weeks CA	Born <30 w GA; living within 100 km of the hospital; English-speaking parents	Congenital anomalies likely to affect neurodevelopment
Spittle et al., 2017 [30]	High/low social risk: 27 w (NI)/27.4 w (NI)	High/low social risk: 27.5 w (NI)/27.4 w (NI)	High/low social risk: 985 g (NI)/1052 g (NI)	High/low social risk: 981 g (NI)/999 g (NI)	2 weeks CA	Born <30 w GA; living within 100 km of the hospital; English-speaking parents	Congenital anomalies likely to affect neurodevelopment
Stedall et al., 2022 [31]	27.5 w (NI)	27.5 w (NI)	1074 g	1037 g	2 weeks CA	Born <30 w GA; living within 100 km of the hospital; English-speaking parents	Congenital anomalies likely to affect neurodevelopment
Van Balen et al., 2019 [32]	<37 w: *n* = 19 ≥37 w: *n* = 2	<37 w: *n* = 23 ≥37 w: *n* = 2	1210 g (585–4750)	1143 g (635–3460)	3 months CA	Infants admitted to NICU with abnormal GMA around 10 w CA	Congenital anomalies; caregivers with inappropriate understanding of the Dutch language
Van Hus et al., 2016 [33]	29.6 w (NI)	30 w (NI)	1242 g (NI)	1306 g (NI)	Just before discharge from NICU	Born <32 w GA and/or <1500 g BW	Severe congenital abnormalities; mother with severe physical or mental illness; no Dutch-speaking parents and absent interpreter; participation in another study on post-discharge management
Youn et al., 2021 [34]	29 w (NI)	29 w (NI)	1145.5 g (NI)	1188.9 g (NI)	3 months CA	Born ≤30 w GA or ≤1500 g BW	Congenital neuromuscular diseases; cardiac anomalies; chromosomal anomalies
Ziegler et al., 2020 [35]	27 w (25–30)	29.5 w (26–31)	850 g (570–1450)	1025 g (690–1400)	Between 35 weeks of gestational age and 4 months of corrected age	Born <32 w; neurological abnormalities indicating a moderate to high risk of CP	Severe congenital disorders; participation in the Erythropoietin for the Repair of Cerebral Injury in Very Preterm Infants study; poor understanding of German of the caregivers

EG: experimental group; CG: control group; w: week; g: gram; h, hour, GA, gestational age; *n*: number; NI: no information, CA: corrected age, PVL: periventricular leukomalacia; IVH: interventricular hemorrhage, HIE: hypoxic–ischemic encephalopathy; BPD: bronchopulmonary dysplasia; NICU: neonatal intensive care unit; HAI: hand assessment for infants; CP: cerebral palsy; TIMP: Test of Infant Motor Performance, MRI: magnetic resonance imaging, ASQ-3: Ages and Stages Questionnaire Third version; pc: percentile; gr, grade; GMA: general movement assessment; GM, general movements; BW: birthweight.

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
