# Peer review of "Early Intervention Including an Active Motor Component in Preterms with Varying Risks for Neuromotor Delay: A Systematic Review and Narrative Synthesis"

_jcm, 2025, doi:10.3390/jcm14041364_

Round 1

Reviewer 1 Report

Comments and Suggestions for Authors

excellent work, well organized, well structured and according to my point of view will be useful in the literature.

I am just wondering if there is any study limitation (eg is there any need for smaller gestational ages?)

Author Response

Comment 1: 

excellent work, well organized, well structured and according to my point of view will be useful in the literature.

I am just wondering if there is any study limitation (eg is there any need for smaller gestational ages?)

Response 1: 

Thank you for your positive feedback and for your insightful question. If I understand your query correctly, you are asking whether there are any study limitations, particularly regarding the inclusion of smaller gestational ages.

We have addressed study limitations in the discussion section of our paper (discussion section, line 367-375). Specifically:

However, this systematic review also has some limitations. At study level, only a few studies included the interaction between group and time in their statistical analyses, which is needed to examine if there are differences between groups over time. Additionally, preterm birth contains still a rather large range or gestational ages with differences in expected outcome and more specific CP-related outcomes were often lacking even in the studies focusing on infants with high risk for CP. Furthermore, specifically for the family-centred interventions, we were not able to extract from the articles the amount of stimulation parents had attributed to the motor development compared to other developmental domains. Hence, this needs to be taken into account when interpreting the results. 

Regarding gestational age, we examined in more detail the range of gestational ages included in the studies. We found that most studies were conducted on mixed populations. Specifically, in 13 studies, the mean or median gestational age fell within the very preterm range (28–31 weeks), while 8 studies included extreme preterm infants (<28 weeks), and 2 studies focused on moderate-to-late preterm infants (32–36 weeks). We have now explicitly included this as a study limitation in our discussion.

We appreciate your valuable comments and hope this clarification addresses your concern.

Reviewer 2 Report

Comments and Suggestions for Authors

Summary:  To augment the Cochrane review on Early Intervention, this systematic review sought to evaluate potential effects of interventions that included active motor components for infants born preterm with and without a high risk for CP.  The authors identified 21 unique (quasi-) RCTs that were reported as either motor-based or family-centered interactions.  They described the different studies in their report but ultimately there was too much heterogeneity to synthesize the results of the studies systematically.  The study is interesting and highlights the current difficulties in creating an evidence base for motor interventions in young infants.  

This report seems to fall between a systematic review and a scoping review.   Although the authors did a great job evaluating the risk of bias in these reports, due to the small numbers and heterogeneity of the existing studies, it was not possible to systematically synthesize the evidence from the studies or provide any definitive conclusions.  Therefore, it might be more appropriate to denote this as a scoping review.  

For a systematic review, please report if the study was pre-registered and if so provide the registration number.  Please provide the appropriate checklist for the study which can be found at prisma-statement.org. (A scoping review checklist is also available in the "Prisma Extensions" tab.) Ensure that all areas of the checklist are addressed, and if not, explain in the paper why this is the case. In my review of the Prisma 2020 checklist, there are some missing elements that the authors should comment on.  

The flow sheet provided in the Prisma website is more clear than the flow chart used by the authors in Figure 1-a suggestion is to transfer the information to this template in the manuscript revision.  

Please provide a summary of the primary outcomes for each study in a new table or add this to Table 2.  Tables 2 and 3 explain the heterogeneity between studies very well but some editing is needed to summarize the information more concisely.  The authors should consider if all the information in the tables is needed.  

The study cohort was described as infants born preterm who were or were not at high risk for CP.  It might be better to categorize the cohort more generally as infants born preterm at (varying) risk for neuromotor delays/impairments. High risk for CP has come to be recognized as a specific designation derived from specific criteria and it's not clear that this designation was used in the studies identified.  Furthermore, developmental tests were the outcome used in many of the studies rather than an outcome of CP.  Focusing the manuscript on the outcome of CP does not seem to match up with the studies presented and the text throughout the paper should be edited to balance this more appropriately.  

In the discussion, the authors should provide a strong statement of what is needed for the field of physical therapy/early intervention to improve the evidence basis for therapy for infants born preterm, based on their findings here. For example, there is a good point to be made that in future studies that evaluate therapies specifically for CP, CP outcomes should be described in addition to developmental assessments.  It would also be beneficial to acknowledge studies published more recently that were not included.  

Author Response

Dear Reviewer,

We sincerely appreciate your thorough review and valuable feedback on our manuscript. Your insightful comments have helped us improve the clarity, accuracy, and overall quality of our work. Below, we provide detailed responses to each of your suggestions and describe the corresponding revisions made in the manuscript.  We believe these changes have strengthened the paper, and hope this addresses your concerns. We thank you for your time and consideration.

Comment 1: This report seems to fall between a systematic review and a scoping review.   Although the authors did a great job evaluating the risk of bias in these reports, due to the small numbers and heterogeneity of the existing studies, it was not possible to systematically synthesize the evidence from the studies or provide any definitive conclusions.  Therefore, it might be more appropriate to denote this as a scoping review.  

Response 1: Thank you very much for your valuable feedback and for taking the time to carefully review our manuscript. We truly appreciate your comments regarding the classification of our review and understand your suggestion to consider it as a scoping review.

However, after careful consideration and reflection on the definitions of both approaches, we believe that our work aligns more closely with a systematic review. Our decision is based on the methodology we have employed in our study, as outlined in the paper by Munn et al. (2018) (https://doi.org/10.1186/s12874-018-0611-x). Munn et al. describe that scoping reviews generally have a broader scope and an overarching purpose, such as mapping the evidence or identifying key concepts and research gaps. By contrast, systematic reviews aim to address a concrete, well-defined research question and often provide implication for practice.

Specifically, in our study:

We followed a predefined protocol, including a comprehensive search strategy and strict inclusion and exclusion criteria.

We conducted a thorough risk of bias assessment, a critical element of systematic reviews, which is less commonly performed in scoping reviews.

We provided a critical synthesis of the available evidence, aiming to offer as much insight as possible into the current body of research, despite the limitations posed by the heterogeneity and small number of studies.

In addition, consistent with the characteristics of systematic reviews described by Munn et al., our review has a concrete and well-defined research question, provides implications for practice by confirming current practices, addressing variation, and identifying new practices, and aims to uncover international evidence to identify conflicting results and inform areas for future research.

While we acknowledge the limitations posed by the small number and heterogeneity of the included studies, these factors do not preclude classification as a systematic review. Systematic reviews inherently aim to summarize the best available evidence, even when the available evidence is limited or diverse.

We hope this explanation clarifies our rationale for classifying the review as systematic, and we remain open to further discussion or suggestions.

Comment 2: For a systematic review, please report if the study was pre-registered and if so provide the registration number.  Please provide the appropriate checklist for the study which can be found at prisma-statement.org. (A scoping review checklist is also available in the "Prisma Extensions" tab.) Ensure that all areas of the checklist are addressed, and if not, explain in the paper why this is the case. In my review of the Prisma 2020 checklist, there are some missing elements that the authors should comment on.  

Response 2: Thank you for your valuable feedback. Unfortunately, this study was not pre-registered. We have now explicitly mentioned this in the Methods and discussion section (lines 68-69 and 388)

We have followed the PRISMA 2020 checklist and included it as a supplementary file. Any missing elements from the checklist are discussed in the limitations section of the discussion (lines 388-389).

Comment 3: The flow sheet provided in the Prisma website is more clear than the flow chart used by the authors in Figure 1-a suggestion is to transfer the information to this template in the manuscript revision.  

Response 3: Thank you for your suggestion. We have now reformatted Figure 1a using the PRISMA flow diagram template to enhance clarity and alignment with PRISMA guidelines.

Comment 4: Please provide a summary of the primary outcomes for each study in a new table or add this to Table 2.  Tables 2 and 3 explain the heterogeneity between studies very well but some editing is needed to summarize the information more concisely.  The authors should consider if all the information in the tables is needed.  

Response 4: Thank you very much for your insightful comments and suggestions. We appreciate your feedback on Tables 2 and 3, and we are pleased that you found they explain the heterogeneity between studies well.

In response to your request for a summary of the primary outcomes for each study, we have made the following adjustments to Table 2:

  • The primary outcome measure of each study is now indicated in bold within the "motor outcome measures" column.
  • If the primary outcome measure was not a motor outcome, or if no primary outcome measure was explicitly mentioned in the study (neither in the paper itself, a registered protocol, nor in any registration we could identify), this has been indicated with a symbol in the "motor outcome measures" column.

Given the heterogeneity of the included studies and our intention to emphasize this diversity, we decided not to make further changes to the tables at this stage. As you noted, these tables already reflect the heterogeneity well, which we consider an important aspect of our findings.

That said, if you have concrete suggestions on how to further improve the tables, we are happy to consider these and implement additional edits where needed.

We hope that these adjustments address your concerns, and we remain open to any additional feedback you may have.

Comment 5: The study cohort was described as infants born preterm who were or were not at high risk for CP.  It might be better to categorize the cohort more generally as infants born preterm at (varying) risk for neuromotor delays/impairments. High risk for CP has come to be recognized as a specific designation derived from specific criteria and it's not clear that this designation was used in the studies identified.  Furthermore, developmental tests were the outcome used in many of the studies rather than an outcome of CP.  Focusing the manuscript on the outcome of CP does not seem to match up with the studies presented and the text throughout the paper should be edited to balance this more appropriately.  

Response 5: Thank you for your insightful feedback. We have revised the text to describe the cohort as "infants born preterm at varying risk of neuromotor delays", as suggested. However, we have retained explicit mentions of CP in certain sections where relevant. Specifically, CP remains included in the search strategy, as its addition significantly increased the number of eligible articles that were not identified using only the terms "preterm" or "neuromotor delay/impairments."

We appreciate your suggestion and have adjusted the manuscript accordingly to better align with the studies presented.

Comment 6: In the discussion, the authors should provide a strong statement of what is needed for the field of physical therapy/early intervention to improve the evidence basis for therapy for infants born preterm, based on their findings here. For example, there is a good point to be made that in future studies that evaluate therapies specifically for CP, CP outcomes should be described in addition to developmental assessments.  It would also be beneficial to acknowledge studies published more recently that were not included. 

Response 6: Thank you for your valuable feedback.In response to your suggestion, we have strengthened the discussion by emphasizing in future research the need for inclusion of  CP-specific outcomes in studies evaluating therapies. (line 363)

Furthermore, we acknowledge that our inclusion criteria (requiring at least 50% of the infants to be preterm or a separate analysis for preterm infants) led to the exclusion of some relevant studies, such as Baby-CIMT (Eliasson et al 2018 and Chamudot et al 2018). We have now explicitly mentioned this limitation in the discussion. (lines 380-385)

Additionally, we recognize that more recent studies, published after our search period, may provide further insights. We have now acknowledged this as a limitation and suggested that future reviews incorporate these newer findings. (lines 380-385).
We appreciate your constructive comments and have revised the manuscript accordingly.

Reviewer 3 Report

Comments and Suggestions for Authors

Early intervention including an active motor component in preterms with or without a high risk for cerebral palsy: a systematic review and narrative synthesis. 

Background: Previous reviews demonstrated stronger benefits of early interventions on cognition compared to motor outcome in preterm born infants. Potentially, motor development needs more targeted interventions, including at least an active motor component. However, there is no overview focusing on such interventions in preterm born infants, despite the increased risk for cerebral palsy. Methods: (quasi-)Randomized controlled trials regarding early interventions in preterm born infants, with or without a high risk for cerebral palsy, were systematically searched in PubMed, Embase and Web of Science, and included if they comprised an active motor component started within the first year. Data of study and participants characteristics were extracted. Risk of bias was assessed with the Risk of Bias 2 tool. Results: Twenty-five studies, including 21 unique (quasi-)RCTs were included and categorized as either pure motor-based interventions (N=6) or family-centred interventions (N=19). Four motor-based interventions improved motor outcomes immediately after the intervention, of which one at follow-up, compared to five and one for family-centred approaches, respectively. Only five family-centred studies assessed long-term effects beyond age five, finding no greater efficacy than standard care. Overall, large variations were present for intervention intensity, type and outcomes between the included studies. Conclusions: Although methodological heterogeneity compromised conclusions, limited effects on motor outcome, in particular long-term outcome, were identified. Including a stronger motor-focused component embedded within a family-centred approach could potentially increase the impact on motor outcome, which would be of particular interest for infants showing early signs of cerebral palsy

This manuscript is very well described and written.

It has been a great pleasure to be able to read it and reviewing it.

Not changes are needed. Only it had been good to include more basedata...but maybe, the results would be the same...since the evidence is limited...

Thank you very much for your work.

Congratulations 

Author Response

Thank you very much for your positive feedback and for taking the time to carefully review our manuscript.

Reviewer 4 Report

Comments and Suggestions for Authors

Dear  Authors, 

Thank you for allowing me to review this article. It is a very interesting review regarding the effectiveness of early interventions with an active motor component started in the first year of life in preterm-born infants with or without the risk of developing cerebral palsy (CP). The article included 25 articles selected after a careful database screening. Four (4) pure motor-based interventions revealed improved motor outcomes from 6 such studies. There were 19 family-centered intervention studies, of which 11 showed similar early improvements. Long-term effects beyond the age of 5 were not obvious, due to several reasons well underlined in the article. The article is well structured. Eligibility criteria are well presented. Study and participant characteristics are clear and presented extensively. There is a well-documented Results area dealing with motor-based and family-centered interventions, which I appreciated. The Discussion section is also very detailed and focused on significant issues that could influence the results/ conclusions,  such as the complex structure of CP and the interaction between motor- and family-centered interventions.  The limitations of the article are well underlined, especially the high level of methodological heterogeneity from the article included in this review.  Of interest is the fact that a meta-analysis would be beneficial to strengthen the conclusions of this article.  

The Conclusions section underlines that early interventions with a motor component are beneficial in preterm newborns with or without a high risk of CP. It is unclear how the interaction between pure motor and family-centered interactions can improve the cognitive and motor outcomes in these babies. I suggest you clarify this part of the article.  Otherwise, the article is well written and worth being published.    

Author Response

Comment 1: 

The Conclusions section underlines that early interventions with a motor component are beneficial in preterm newborns with or without a high risk of CP. It is unclear how the interaction between pure motor and family-centered interactions can improve the cognitive and motor outcomes in these babies. I suggest you clarify this part of the article.  Otherwise, the article is well written and worth being published.    

Response 1: 

Thank you for your feedback and for your positive assessment of our work.

We acknowledge that cognitive outcomes were mentioned in the text, even though assessing the effect of interventions on cognitive function was beyond the scope of this review. So we could not draw any conclusions on this domain.

To avoid any further confusion, we have now removed references to cognitive outcomes from the conclusion section. 

This systematic review provided an overview of the evidence on the effectiveness of early interventions with an active motor component in preterm born infants withvarying risk for neuromotor delays, which were classified as either pure motor-based interventions or family-centred interventions. Although methodological heterogeneity between studies compromised study conclusions and clinical implications, we identified limited effects of these early interventions on motor outcome in particular for long-term follow-up. Acknowledging the importance of family-centred interventions for other outcomes, we hypothesize that including a stronger motor-focused component within a family-centred approach would increase the impact on motor outcome, which would be of particular interest for infants at high risk for CP.

We appreciate your careful review and constructive suggestions.

Reviewer 5 Report

Comments and Suggestions for Authors

The manuscript entitled "Early Intervention Including an Active Motor Component in Preterm Infants with or without a High Risk for Cerebral Palsy: A Systematic Review and Narrative Synthesis" constitutes a meticulously organized and perceptive addition to the domain of neonatal rehabilitation. It offers an exhaustive examination of early interventions aimed at enhancing motor outcomes in preterm infants, effectively consolidating findings from 25 distinct studies. The authors have diligently adhered to PRISMA guidelines and utilized suitable methodologies, including a thorough assessment of risk of bias, thereby augmenting the credibility of their conclusions. The distinction made between motor-based and family-centered interventions, along with their corresponding outcomes, represents a significant strength, providing clear implications for clinical practice and prospective research endeavors. The study underscores the necessity of integrating targeted motor components within family-centered approaches to maximize developmental outcomes, which is both pragmatic and innovative. The manuscript is articulated with precision, scientifically robust, and offers substantive insights into a vital sector of pediatric care.

Author Response

Comment 1: 

The manuscript is articulated with precision, scientifically robust, and offers substantive insights into a vital sector of pediatric care.

Thank you very much for your positive feedback and for taking the time to carefully review our manuscript.